# Evaluation of polygenic prediction methodology within a reference-standardized framework

**Oliver Pain** [1,2]\*, **Kylie P. Glanville** [1], **Saskia P. Hagenaars** [1], **Saskia Selzam** [1], **Anna E. Fürtjes** [1], **Héléna A. Gaspar** [1], **Jonathan R. I. Coleman** [1], **Kaili Rimfeld** [1], **Gerome Breen** [1,2], **Robert Plomin** [1], **Lasse Folkersen** [3], **Cathryn M. Lewis** [1,2,4]

**1** Social, Genetic and Developmental Psychiatry Centre, Institute of Psychiatry, Psychology and Neuroscience, King's College London, London, United Kingdom, **2** NIHR Maudsley Biomedical Research Centre, South London and Maudsley NHS Trust, London, United Kingdom, **3** Institute of Biological Psychiatry, Sankt Hans Hospital, Copenhagen, Denmark, **4** Department of Medical and Molecular Genetics, Faculty of Life Sciences and Medicine, King's College London, London, United Kingdom

\* oliver.pain@kcl.ac.uk

## Abstract

The predictive utility of polygenic scores is increasing, and many polygenic scoring methods are available, but it is unclear which method performs best. This study evaluates the predictive utility of polygenic scoring methods within a reference-standardized framework, which uses a common set of variants and reference-based estimates of linkage disequilibrium and allele frequencies to construct scores. Eight polygenic score methods were tested: p-value thresholding and clumping (pT+clump), SBLUP, lassosum, LDpred1, LDpred2, PRScs, DBSLMM and SBayesR, evaluating their performance to predict outcomes in UK Biobank and the Twins Early Development Study (TEDS). Strategies to identify optimal p-value thresholds and shrinkage parameters were compared, including 10-fold cross validation, pseudovalidation and infinitesimal models (with no validation sample), and multi-polygenic score elastic net models. LDpred2, lassosum and PRScs performed strongly using 10-fold cross-validation to identify the most predictive p-value threshold or shrinkage parameter, giving a relative improvement of 16–18% over pT+clump in the correlation between observed and predicted outcome values. Using pseudovalidation, the best methods were PRScs, DBSLMM and SBayesR. PRScs pseudovalidation was only 3% worse than the best polygenic score identified by 10-fold cross validation. Elastic net models containing polygenic scores based on a range of parameters consistently improved prediction over any single polygenic score. Within a reference-standardized framework, the best polygenic prediction was achieved using LDpred2, lassosum and PRScs, modeling multiple polygenic scores derived using multiple parameters. This study will help researchers performing polygenic score studies to select the most powerful and predictive analysis methods.

**Data Availability Statement:** A data transfer agreement is required to access individual-level data for TEDS (https://www.teds.ac.uk/researchers/teds-data-access-policy) and UK

Biobank(https://biobank.ctsu.ox.ac.uk/crystal/exinfo.cgi?src=accessing_data_guide). The code used during this study are available at GitHub: https://opain.github.io/GenoPred.

**Funding:** This study was funded by the UK Medical Research Council (https://mrc.ukri.org/): MR/N015746/1 to CML and MR/S0151132 to SH. CML was also funded by the National Institute for Health Research (NIHR) Biomedical Research Centre at South London and Maudsley NHS Foundation Trust and King's College London (https://www.maudsleybrc.nihr.ac.uk/). The funders had no role in study design, data collection and analysis, decision to publish, or preparation of the manuscript.

**Competing interests:** I have read the journal's policy and the authors of this manuscript have the following competing interests: Cathryn Lewis sits on the Myriad Neuroscience Scientific Advisory Board. The other authors declare no competing interests.

## Author summary

An individual's genetic predisposition to a given outcome can be summarized using polygenic scores. Polygenic scores are widely used in research and could also be used in a clinical setting to enhance personalized medicine. A range of methods have been developed for calculating polygenic scores, but it is unclear which methods are the best. Several methods provide multiple polygenic scores for each individual which must then be tested in an independent tuning sample to identify which polygenic score is most accurate. Other methods provide a single polygenic score and therefore do not require a tuning sample. Our study compares the prediction accuracy of eight leading polygenic scoring methods in a range of contexts. For methods that calculate multiple polygenic scores, we find that LDpred2, lassosum, and PRScs methods perform best on average. For methods that provide a single polygenic score, not requiring a tuning sample, we find PRScs performs best, and the faster DBSLMM and SBayesR methods also perform well. Our study has provided a comprehensive comparison of polygenic scoring methods that will guide future implementation of polygenic scores in both research and clinical settings.

## Introduction

In personalized medicine, medical care is tailored for the individual to provide improved disease prevention, prognosis, and treatment. Genetics is a potentially powerful tool for providing personalized medicine as genetic variation accounts for a large proportion of individual differences in health and disease [1]. Furthermore, an individual's genetic sequence is stable across the lifespan, enabling predictions long before the onset of most diseases. Although genetic information is used to predict rare Mendelian genetic disorders, such as breast cancer based on *BRCA1/2* variants, our ability to predict common disorders using genetic information is currently insufficient for clinical implementation. This is due to the increased etiological complexity of common disorders, with complex interplay between genetic and environmental factors, and the highly polygenic genetic architecture with contributions from many genetic variants with small effect sizes [2]. However, genome-wide association studies (GWAS), used to detect common genetic associations, are rapidly increasing in sample size, and are identifying large numbers of novel and robust genetic associations for health-related outcomes [3]. This growing source of information is also improving our ability to predict an individual's disease risk or measured trait based on their genetic variation [4,5].

An individual's genetic risk for an outcome can be summarized in a polygenic score, calculated from the number of trait-associated alleles carried. The contributing variants are typically weighted by the magnitude of effect they confer on the outcome of interest, estimated in a reference GWAS. There are several challenges in performing a well-powered polygenic score analysis. Firstly, GWAS effect-sizes are inflated through Winner's curse, and unbiased estimates can only be obtained through an independent training sample, with these effect-size estimates then used to calculate polygenic scores in a further independent sample [6]. Secondly, to maximize polygenic prediction accuracy, the GWAS summary statistics must be adjusted to account for the linkage disequilibrium (LD) between genetic variants, to avoid double counting the non-independent effect of variants in high LD, and account for varying degrees of polygenicity across outcomes, i.e. the number of genetic variants affecting the outcome [6]. LD can be accounted for using LD-based clumping of GWAS summary statistics, removing variants in LD with lead variants within each locus, and polygenicity is accounted for by applying multiple GWAS $p$-value thresholds (pT) to select the effect alleles included in

the polygenic score [4,5]. This pT+clump approach is conceptually simple and computationally scalable [7]. However, using a hard LD threshold in clumping to retain or remove variants from the polygenic score calculation can potentially reduce the variance explained by the polygenic score. Alternative summary statistic-based polygenic score methods retain all genetic variants by modelling both the LD between variants and the polygenicity of the outcome [8–14]. These methods use estimates of LD to jointly estimate the effect of nearby genetic variation maximizing the signal captured, and generally apply a shrinkage parameter to the genetic effects to reduce overfitting and allow for varying degrees of polygenicity across outcomes.

Polygenic scoring methods can lead to overfitting of genetic effects due to the p-value based selection of variants or joint estimation of many genetic effects. To avoid this overfitting, genetic effect size estimates can be reduced using shrinkage methods to improve the generalizability of the model. Shrinkage methods for polygenic scoring can be separated into frequentist penalty-based methods (e.g. lasso regression-based lassosum [10], summary-based best linear unbiased prediction (SBLUP) [9]) and Bayesian methods that shrink estimates to fit a prior distribution of effect sizes, such as LDpred1 [8], LDpred2 [13], PRScs [11], SBayesR [12], and DBSLMM [14]. Each of these methods have been shown to improve the predictive utility of polygenic scores over those derived using the pT+clump approach. In comparisons between methods the findings are mixed: some studies have similar results across methods [15], while papers developing a new method often report that the developed method out-performs chosen other methods. To our knowledge no independent study has yet compared all approaches.

Five methods (pT+clump, LDpred1, LDpred2, lassosum and PRScs) generate multiple polygenic scores from user-defined tuning parameters. To determine which tuning parameter provides optimal prediction, the polygenic scores must first be tested in an independent 'tuning' sample. The pT+clump approach applies p-value thresholds to select variants included in the polygenic score, whereas LDpred1, LDpred2, lassosum and PRScs apply shrinkage parameters to adjust the GWAS effect sizes. In addition, lassosum, PRScs and LDpred2 provide a pseudo-validation approach, whereby a single optimal shrinkage parameter is estimated based on the GWAS summary statistics alone, and therefore do not require a tuning sample. SBayesR and DBSLMM can be considered pseudovalidation approaches as they also do not require a tuning sample to identify optimal parameters. Another approach to derive polygenic scores is to assume an infinitesimal model, as is done by SBLUP and the infinitesimal models of LDpred1 and 2 [16]. Similar to pseudovalidation approaches, no tuning sample is required when assuming an infinitesimal model. Rather than selecting a single tuning parameter, some studies have suggested that combining polygenic scores across p-value thresholds whilst taking into account their correlation using either PCA or model stacking can improve prediction [17,18].

Polygenic scores are a useful research tool, as well as a promising potential tool for personalized healthcare through prediction of disease risk, prognosis, and treatment response [19]. However, polygenic scores calculated in a clinical setting should be valid for a single target sample and thus need to be constructed using a reference-standardized framework. Here, the polygenic score is independent of any properties specific to the target sample, including the genetic variation available, and the LD and minor allele frequency (MAF) estimates. In a reference-standardized approach, the genetic variants considered can be standardized by using only single nucleotide polymorphisms (SNPs) that are commonly available after imputation, such as variation within the HapMap3 reference [20]. The LD and MAF estimates can be standardized by using an ancestry matched individual-level genetic dataset such as 1000 Genomes [21]. Determining these properties (SNPs, LD, MAF) in reference data provides a practical approach for estimating polygenic scores for an individual, making them comparable to polygenic scores for other individuals of the same ancestry [22]. Use of a reference-standardized framework also offers advantages by improving the comparability of polygenic scores across cohorts. Several polygenic scoring methods now

recommend the use of HapMap3 SNPs and precomputed external LD estimate references [11–13], in line with a reference-standardized approach.

In this study, we perform an extensive comparison of polygenic scoring methods within a reference-standardized framework. We evaluate the predictive utility of models for outcomes in UK Biobank (UKB) and the Twins Early Development Study (TEDS), combining information across tuning parameters. We evaluate eight polygenic scoring methods and apply different modelling strategies to select optimal tuning parameters to establish the combinations that perform consistently well. The reference-standardized framework increases the generalizability of results and provides a resource for future studies investigating polygenic prediction in a research study or clinical setting.

## Methods

To evaluate the different polygenic scoring approaches, we used two target samples: UK Biobank (UKB) [23], and the Twins Early Development Study (TEDS) [24]. All code used to prepare data and carryout analyses is available on the GenoPred website (see Data and Code Availability).

### Ethics statement

For UKB, the protocol and written consent were approved by the UKB's Research Ethics Committee (Ref: 11/NW/0382). For TEDS, ethical approval for TEDS has been provided by the King's College London ethics committee (reference: 05/Q0706/228), with written parental and/or self-consent obtained before data collection.

### UKB

UKB is a prospective cohort study that recruited >500,000 individuals aged between 40–69 years across the United Kingdom.

**Genetic data.** UKB released imputed dosage data for 488,377 individuals and ~96 million variants, generated using IMPUTE4 software [23] with the Haplotype Reference Consortium reference panel [25] and the UK10K Consortium reference panel [26]. This study retained individuals that were of European ancestry based on 4-means clustering on the first 2 principal components provided by the UKB (self-reported ancestry was not used), and removed related individuals (>3rd degree relative) using relatedness kinship (KING) estimates provided by the UKB [23]. The imputed dosages were converted to hard-call format using a hard call threshold of zero.

**Phenotype data.** Eleven UKB phenotypes were analyzed. Eight phenotypes were binary: Depression, Type II Diabetes (T2D), Coronary Artery Disease (CAD), Inflammatory Bowel Disease (IBD), Rheumatoid arthritis (RheuArth), Multiple Sclerosis (MultiScler), Breast Cancer, and Prostate Cancer. Three phenotypes were continuous: Intelligence, Height, and Body Mass Index (BMI). Further information regarding outcome definitions can be found in S1 Text.

Analysis was performed on a subset of ~50,000 UKB participants for each outcome. For each continuous trait (Intelligence, Height, BMI), a random sample was selected. For disease traits, all cases were included, except for Depression and CAD where a random sample of 25,000 cases was selected. Controls were randomly selected to obtain a total sample size of 50,000. Sample sizes for each phenotype after genotype data quality control are shown in Table 1. S1 Fig shows a schematic diagram of how UKB data was split into training and testing samples.

### TEDS

TEDS is a population-based longitudinal study of twins born in England and Wales between 1994 and 1996 [27]. For this study, one individual from each twin pair was removed to retain only unrelated individuals.

**Table 1. Sample size of target sample phenotypes after quality control.**

| UKB Phenotype | Description | Total sample size | No. of cases | No. of controls |
|---|---|---|---|---|
| Depression | Major depression | 50000 | 25000 | 25000 |
| Intelligence | Fluid intelligence | 50000 | NA | NA |
| BMI | Body Mass Index | 50000 | NA | NA |
| Height | Height | 50000 | NA | NA |
| T2D | Type-2 Diabetes | 50000 | 35112 | 14888 |
| CAD | Coronary Artery Disease | 50000 | 25000 | 25000 |
| IBD | Inflammatory Bowel Disease | 50000 | 46539 | 3461 |
| MultiScler | Multiple Sclerosis | 50000 | 48863 | 1137 |
| RheuArth | Rheumatoid Arthritis | 50000 | 46592 | 3408 |
| Prostate Cancer | Prostate Cancer | 50000 | 47073 | 2927 |
| Breast Cancer | Breast Cancer | 50000 | 41488 | 8512 |
| **TEDS Phenotype** | | | | |
| GCSE | Mean GCSE scores | 7296 | NA | NA |
| ADHD | ADHD symptoms | 7880 | NA | NA |
| BMI21 | Body Mass Index at age 21 | 5220 | NA | NA |
| Height21 | Height at age 21 | 5455 | NA | NA |

**Genetic data.** TEDS participants were genotyped using two arrays, HumanOmniExpressExome-8v1.2 and AffymetrixGeneChip 6.0. Stringent quality control was performed separately for each array, prior to imputation via the Sanger Imputation server using the Haplotype Reference Consortium (release 1.1) reference data [25,28]. Imputed genotype dosages were converted to hard-call format using a hard call threshold of 0.9, with variants for each individual set to missing if no genotype had a probability of >0.9. Variants with an INFO score <0.4, MAF <0.001, missingness >0.05 or Hardy-Weinberg equilibrium p-value <$1\times10^{-6}$ were removed.

**Phenotypic data.** This study used four continuous phenotypes within TEDS: Height, Body Mass Index (BMI), Educational Achievement, and Attention Deficit Hyperactivity Disorder (ADHD) symptom score (Table 1). These phenotypes were selected based on a previous polygenic study, enabling comparison across methods [29]. The phenotypes were derived using the same protocol as previously.

## Genotype-based scoring

The following genotype-based scoring procedure provides reference standardized polygenic scores and can be applied to any datasets of imputed genome-wide array data (Fig 1).

**SNP-level QC.** HapMap3 variants from the LD-score regression website (see Web Resources) were extracted from target samples (UKB, TEDS), inserting any HapMap3 variants that were not available in the target sample as missing genotypes (as required for reference MAF imputation by the PLINK allelic scoring function) [30]. No other SNP-level QC was performed.

**Individual-level QC.** Individual-level QC prior to imputation was previously performed for both UKB [23] and TEDS [28] samples. Only individuals of European ancestry were retained for polygenic score analysis. They were identified using 1000 Genomes Phase 3 projected principal components of population structure, retaining only those within three standard deviations from the mean for the top 100 principal components. This process will also remove individuals who are outliers due to technical genotyping or imputation errors.

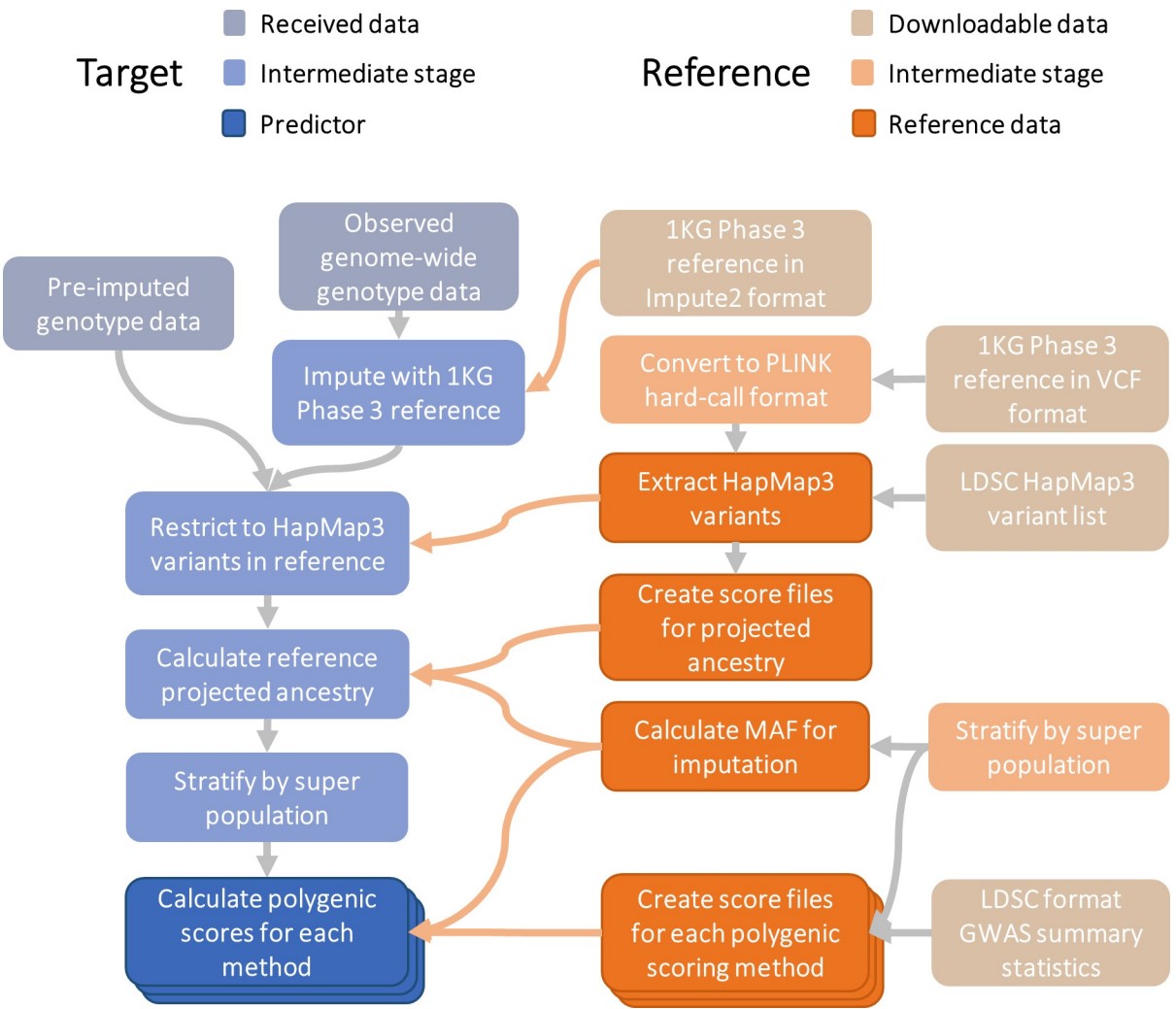

**Fig 1. Schematic diagram of reference-standardized polygenic scoring.** 1KG = 1000 Genomes; LDSC = Linkage Disequilibrium Score Regression; MAF = Minor allele Frequency; Pre-imputed genotype data = Indicates the observed genotype data has already been imputed; Observed genome-wide genotype data = Indicate the observed genotype data has not been imputed, and therefore requires imputation.

**GWAS summary statistics.** GWAS summary statistics were identified for phenotypes the same as or similar as possible to the UKB and TEDS phenotypes (descriptive statistics in S1 Table), excluding GWAS with documented sample overlap with the target samples. GWAS summary statistics underwent quality control to extract HapMap3 variants, remove ambiguous variants, remove variants with missing data, flip variants to match the reference, retain variants with a minor allele frequency (MAF) >0.01 in the European subset of 1KG Phase 3, retain variants with a MAF >0.01 in the GWAS sample (if available), retain variants with a INFO >0.6 (if available), remove variants with a discordant MAF (>0.2) between the reference and GWAS sample (if available), remove variants with p-values >1 or ≤ 0, remove duplicate variants, remove variants with sample size >3SD from the median sample size (if per variant sample size is available).

**Reference genotype datasets.** Target sample genotype-based scoring was performed using two different reference genotype datasets, the European subset of 1000 Genomes Phase 3 (N = 503) and a random subset of 10,000 European-ancestry UKB participants. The UKB

**Table 2. Description of polygenic scoring approaches.**

| Method | Multiple tuning parameters | Pseudo-validation/ infinitesimal option | Software | Description | Parameters | MHC region | LD-reference |
|---|---|---|---|---|---|---|---|
| pT+clump [30] | Yes | No | PLINK | LD-based clumping and p-value thresholding | 10 nested p-value thresholds: 1e-8, 1e-6, 1e-4, 1e-2, 0.1, 0.2, 0.3, 0.4, 0.5, 1 Clumping: $r^2$ = 0.1; window = 250kb | Only top variant retained | EUR 1KG, EUR 10K UKB |
| lassosum [10] | Yes | Pseudo-validation | lassosum | Lasso regression-based | 80 s and lambda combinations: s = 0.2, 0.5, 0.9, 1. lambda = exp(seq(log (0.001), log(0.1), length.out = 20))[A] | Not excluded | EUR 1KG, EUR 10K UKB |
| PRScs[11] | Yes | Pseudo-validation | PRScs | Bayesian shrinkage | 5 global shrinkage parameters (phi) = 1e-6, 1e-4, 1e-2, 1, auto | Not excluded | PRScs-provided EUR 1KG |
| SBLUP[9] | No | Infinitesimal (only option) | GCTA | Best Linear Unbiased Prediction | NA | Not excluded | EUR 1KG, EUR 10K UKB |
| SBayesR [12] | No | Pseudo-validation (only option) | GCTB | Bayesian shrinkage | NA | Excluded (as recommended) | EUR 1KG, EUR 10K UKB, GCTB-provided |
| LDpred1 [8] | Yes | Infinitesimal | LDpred | Bayesian shrinkage | Infinitesimal model and 7 non-zero effect fractions (p) = 3e-3, 1e-3, 3e-2, 1e-2, 3e-1, 1e-1, 1 | Not excluded | EUR 1KG, EUR 10K UKB |
| LDpred2 [13] | Yes | Pseudo-validation and infinitesimal | bigsnpr | Bayesian Shrinkage | Auto, infinitesimal, and grid modes. Grid includes 126 combinations of heritability and non-zero effect fractions (p). | Not excluded | EUR 1KG, EUR 10K UKB |
| DBSLMM | No | Yes (only option) | DBSLMM | Bayesian shrinkage | NA | Not excluded | EUR 1KG, EUR 10K UKB |

*Note.* Default or recommended parameters were used for all methods.

[A] lassosum lambda values described using R code.

reference set was independent of the target sample used for evaluating polygenic scoring methods. These references were used to determine whether the sample size of the reference genotype dataset affects the prediction accuracy of polygenic scores. Only 1,042,377 HapMap3 variants were available in the UKB dataset and used in genotype-based scoring.

**Polygenic Scores (PRS).** Polygenic scoring was carried out using eight approaches with default parameters outlined in Table 2. To ensure comparability across methods, the same set of HapMap3 variants were considered, and the same reference genotype datasets were used to estimate LD and MAF (except for PRScs and SBayesR).

PRScs-provides an LD reference for HapMap3 variants based on the European subset of the 1000 Genomes, and results should be comparable to other methods when using the 1000 Genomes reference. PRScs was not applied using the larger UKB reference dataset as PRScs has been previously reported to show minimal improvement when using larger LD reference datasets [11].

SBayesR analysis requires shrunk and sparse LD matrices as input. LD matrices were calculated using Genome-wide Complex Trait Bayesian analysis (GCTB) [31] in batches of 5,000 variants, which were then merged for each chromosome, shrunk, and then made sparse. SBayesR analysis was also performed using LD matrices released by the developers of GCTB based on 50,000 European UKB individuals (see Web Resources).

Two additional modifications of the standard pT+clump approach were tested, termed 'pT +clump (non-nested)' and 'pT+clump (dense)'. The pT+clump (non-nested) approach is the same the standard pT+clump approach except non-overlapping p-value thresholds were used to select variants included in the polygenic score, thereby making the polygenic scores for each threshold independent. The pT+clump (dense) approach is the same as the standard pT

+clump approach except that it uses 10,000 p-value thresholds (minimum = $5\times10^{-8}$, maximum = 0.5, interval = 5×10–5), implemented using default settings in PRSice [7].

After adjustment of GWAS summary statistics as necessary for each polygenic scoring method, polygenic scores were calculated using PLINK with reference MAF imputation of missing data. All scores were standardized based on the mean and standard deviation of polygenic scores in the reference sample.

To determine whether certain methods are more prone to capturing genetic effects driven by population stratification, we carried out a sensitivity analysis, in which the first 20 principal components were regressed from the polygenic scores in advance. Principal components were derived in the 1KG Phase 3 reference, and then projected into UKB and TEDS samples.

**Modelling approaches.** For methods that provide polygenic scores based on a range of p-value thresholds (pT+clump) or shrinkage parameters (lassosum, PRScs, LDpred1, LDpred2), the best parameter was identified using either 10-fold cross validation (10FCVal) and, if available, pseudovalidation (PseudoVal). Pseudovalidation was performed using the pseudovalidate function in lassosum, the fully-Bayesian approach in PRScs, the auto model in LDpred2. SBayesR and DBSLMM by default estimate the optimal parameters and are therefore considered pseudovalidation methods. Methods assuming an infinitesimal model were SBLUP and the infinitesimal models of LDpred1 and 2. In addition to selecting the single 'best' parameter for polygenic scoring, elastic net models were derived containing polygenic scores based on a range of parameters for each method, with elastic net shrinkage parameters derived using 10-fold cross-validation (Multi-PRS). The number of scores generated by each method, which were included in the multi-PRS model, are shown in Table 2. In addition, we tested whether combining polygenic scores from all methods in an elastic net model improved prediction. This combined model is referred to the 'All' model.

The optimal parameters (pT, GWAS-effect size shrinkage, elastic net parameters) were determined based on the largest mean correlation between observed and predicted values obtained through 10-fold cross validation, and the resulting model was then applied to an independent test set. Ten-fold cross-validation is liable to overfitting when using penalized regression as hyperparameters are tuned using the 10-fold cross validation procedure. The independent test-set validation avoids any overfitting as the independent test sample is not used for hyperparameter tuning. Ten-fold cross validation was performed using 80% of the sample and the remaining 20% was used as the independent test sample. Ten-fold cross validation and test-set validation was carried out using the 'caret' R package, setting the same random seeds prior to subsetting individuals to ensure the same individuals were included for all polygenic scoring methods.

**Evaluating prediction accuracy.** Prediction accuracy was evaluated as the Pearson correlation between the observed and predicted outcome values. Correlation was used as the main test statistic as it is applicable for both binary and continuous outcomes and standard errors are easily computed as

$$SE_r = \frac{1 - r^2}{\sqrt{n - 2}} \tag{1}$$

Where $SE_r$ is the standard error of the Pearson correlation, $r$ is the Pearson correlation, and $n$ is the sample size. Correlations can be easily converted to other test statistics such as $R^2$ (observed or liability) and area under the curve (AUC) (equations 8 and 11 in [32]), with relative performance of each method remaining unchanged.

When modelling the polygenic scores, logistic regression was used for predicting binary outcomes, and linear regression was used for predicting continuous outcomes. If the model

contained only one predictor, a generalized linear model was used. If the model contained more than one predictor (i.e. the polygenic scores for each p-value threshold or shrinkage parameter), an elastic net model was applied to avoid overfitting due to the inclusion of multiple correlated predictors [33].

The correlation between observed and predicted values of each model were compared using William's test (also known as the Hotelling-Williams test) [34] as implemented by the 'psych' R package's 'paired.r' function, with the correlation between model predictions of each method specified to account for their non-independence. A two-sided test was used when calculating p-values.

The correlation between predicted and observed values were combined across phenotypes for each polygenic score method. Correlations and their variances ($SE^2$) were aggregated using the 'BHHR' method [35] as implemented in the 'MAd' R package's 'agg' function, using a phenotypic correlation matrix to account for the non-independence of analyses within each target sample. In addition to averaging results across all phenotypes, we estimate the average performance of methods within high and low polygenicity phenotypes. The polygenicity of phenotypes was estimated using AVENGEME [36] (more information in S1 Text).

The percentage difference between methods was calculated as

$$\% \ difference = ((r_1 - r_2)/r_2) * 100 \tag{2}$$

Where $r_1$ and $r_2$ indicate the Pearson correlation between predicted and observed values for models 1 and 2, respectively.

**Method runtime comparison.** To compare the time taken for each polygenic scoring method to process GWAS summary statistics, we ran each method using GWAS summary statistics restricted to variants on chromosome 22. No parallel implementations were used in this comparison. Apart from LDpred1, all the polygenic scoring methods can be implemented in parallel.

## Results

The eight polygenic risk score methods were applied to the target datasets of UKB (11 phenotypes) and TEDS (4 phenotypes), using two reference data sets of 1000 Genomes (1KG, 503 individuals) and UKB (10,000 individuals). Models were derived using 10-fold cross-validation, pseudovalidation, infinitesimal PRS and analysis of multiple threshold PRS, as appropriate for each polygenic risk score method (Table 2).

First, we confirmed that the design of the study was appropriate to detect differences between the methods using the GWAS summary statistics and test data sets chosen. GWAS summary statistics had sample sizes of a mean of 50,698 cases and 94,391 controls, and 423,698 individuals for continuous traits, with heritability on the liability scale (estimated from the GWAS) ranging between 0.021 (Multiple Sclerosis) and 0.542 for Crohn's disease (S1 Table). For pT+clump, with 1KG reference and UKB target samples, the correlations between observed values (case-control status or measured trait) and the predicted values from the polygenic risk scoring models ranged from 0.074 (SE = 0.010) for Intelligence to 0.299 (SE = 0.010) for Height (S7 Table). For each disorder or trait, reference panel and polygenic scoring method, the correlation was significantly different from zero (S6–S9 Tables). These results confirm that the study design—comprising the GWAS, reference panel, target studies and traits—had sufficient information to capture polygenic prediction, and that the traits are diverse in polygenic architecture.

Results were highly concordant across the different target and reference samples used though the estimates were more precise when using the UKB target sample due to the increased sample size compared to TEDS (S2 and S3 Figs).

**Table 3. Average test-set correlation between predicted and observed values across phenotypes.**

| Method | Model | CrossVal R (SE) | IndepVal R (SE) |
|---|---|---|---|
| pT+clump | 10FCVal | 0.155 (0.002) | 0.153 (0.004) |
| pT+clump | MultiPRS | 0.175 (0.002) | 0.174 (0.004) |
| lassosum | 10FCVal | 0.19 (0.002) | 0.183 (0.004) |
| lassosum | MultiPRS | 0.199 (0.002) | 0.194 (0.004) |
| lassosum | PseudoVal | 0.159 (0.002) | 0.157 (0.004) |
| PRScs | 10FCVal | 0.19 (0.002) | 0.183 (0.004) |
| PRScs | MultiPRS | 0.194 (0.002) | 0.187 (0.004) |
| PRScs | PseudoVal | 0.188 (0.002) | 0.182 (0.004) |
| SBLUP | Inf | 0.162 (0.002) | 0.156 (0.004) |
| SBayesR | PseudoVal | 0.17 (0.002) | 0.167 (0.004) |
| LDpred1 | 10FCVal | 0.178 (0.002) | 0.171 (0.004) |
| LDpred1 | MultiPRS | 0.181 (0.002) | 0.175 (0.004) |
| LDpred1 | Inf | 0.163 (0.002) | 0.156 (0.004) |
| LDpred2 | 10FCVal | 0.194 (0.002) | 0.187 (0.004) |
| LDpred2 | MultiPRS | 0.197 (0.002) | 0.191 (0.004) |
| LDpred2 | PseudoVal | 0.155 (0.002) | 0.151 (0.004) |
| LDpred2 | Inf | 0.161 (0.002) | 0.155 (0.004) |
| DBSLMM | PseudoVal | 0.182 (0.002) | 0.175 (0.004) |
| All | MultiPRS | 0.202 (0.002) | 0.197 (0.004) |

*Note*. This table shows results based on the UKB target sample and 1000 genomes reference. 10FCVal = Single polygenic score based on the optimal parameter as identified using 10-fold cross-validation. Multi-PRS = Elastic net model containing polygenic scores based on a range of parameters, with elastic net shrinkage parameters derived using 10-fold cross-validation. PseudoVal = Single polygenic score based on the predicted optimal parameter as identified using pseudovalidation, which requires no tuning sample, Inf = Single polygenic score based on infinitesimal model, which requires no tuning sample.

### Effect of reference panel and validation method

Polygenic scoring methods were applied to two reference panels of European ancestry: 503 individuals from the 1,000 Genomes sample, and 10,000 individuals from UKB. On average, results were highly similar for both panels (S2 and S3 Figs). For example, with the larger reference panel the correlation increased by a mean of 0.002 in UKB, and 0.008 in TEDS, across traits and polygenic scoring methods (test-set validation, S2–S5 Tables; excluding PRScs which used only the 1,000 Genomes reference panel). The greatest improvements with the larger reference panel were for SBayesR and LDpred2 pseudovalidation methods, with an average increase in correlation of 0.011 and 0.017 respectively. Detailed results are reported here only for the 1,000 Genomes (1KG) reference panel, with full results for UKB reference panel in S1–S20 Tables and S1–13 Figs.

Both 10-fold cross validation and test-set validation methods were used in modelling, across all polygenic risk scoring methods. The 10-fold cross validation results were highly congruent with test-set validation results (Table 3). Results reported are based on test-set validation since this method is clearly robust to overfitting when using elastic net models (see S1–S20 Tables for 10-fold cross-validation results).

### Overview of polygenic scoring methods by modelling strategy

The performance for each polygenic scoring method across phenotypes was assessed using the correlation between observed and fitted values (Fig 2A), and then comparing each method with a baseline method of pT+clump with 10-fold cross validation using the difference in

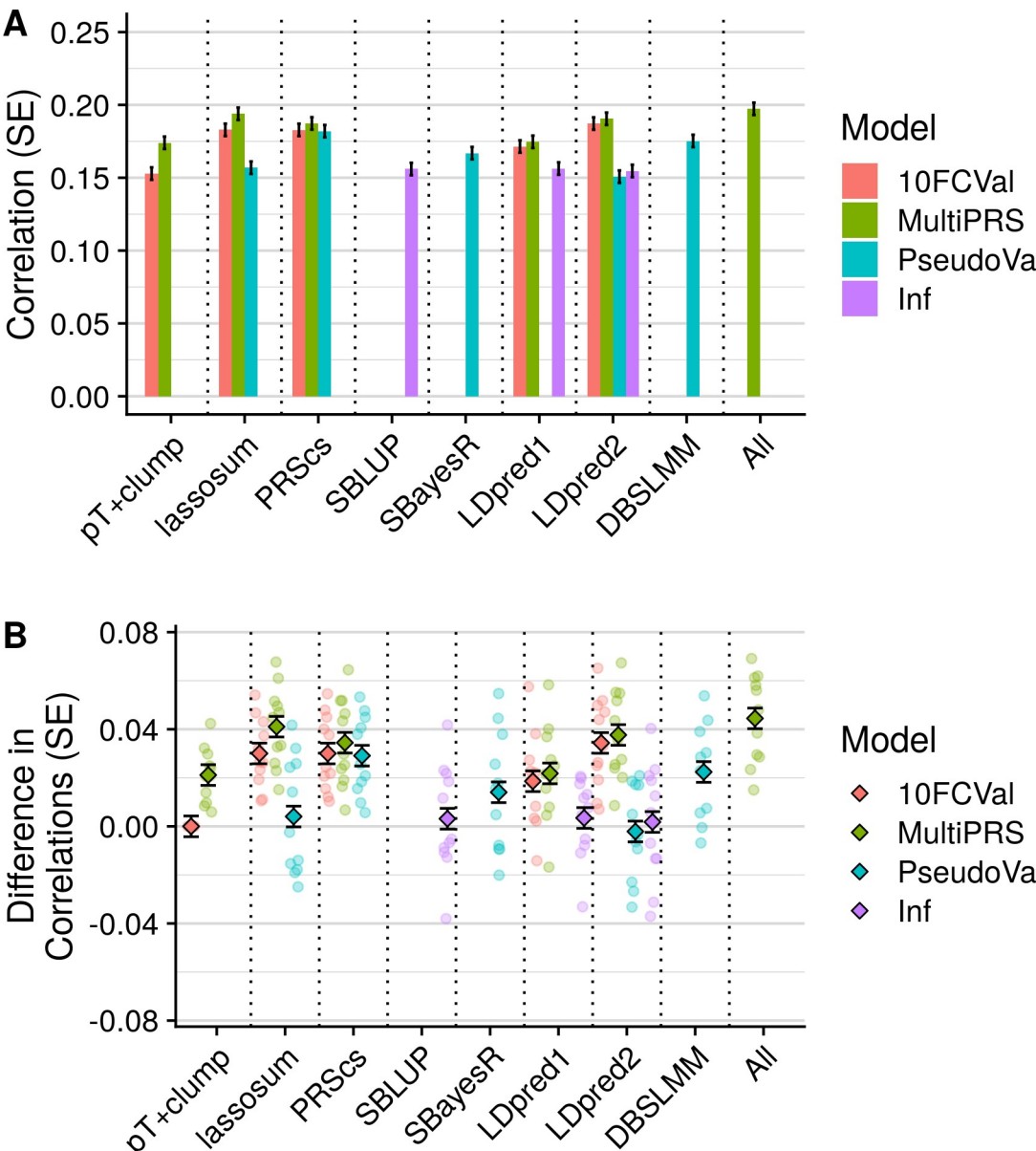

**Fig 2. Polygenic scoring methods comparison for UKB target sample with 1KG reference.** A) Average test-set correlation between predicted and observed values across phenotypes. B) Average difference between observed-prediction correlations for the best pT+clump polygenic score and all other methods. The average difference across phenotypes are shown as diamonds and the difference for each phenotype shown as transparent circles. Shows only results based on the UKB target sample when using the 1KG reference. Error bars indicate standard error of correlations for each method. 10FCVal represents a single polygenic score based on the optimal parameter as identified using 10-fold cross-validation. Multi-PRS represents an elastic net model containing polygenic scores based on a range of parameters, with elastic net shrinkage parameters derived using 10-fold cross-validation. PseudoVal represents a single polygenic score based on the predicted optimal parameter as identified using pseudovalidation, which requires no tuning sample. Inf represents a single polygenic score based on the infinitesimal model, which requires no tuning sample.

correlation (Fig 2B). All methods performed at least as well as pT+clump. These overview results show that the pseudovalidation (PseudoVal) and infinitesimal models (Inf) performed less well than polygenic scores selected through 10-fold cross-validation (10FCVal), and that the prediction when modelling multiple PRS (multi-PRS) was slightly higher than the 10-fold

cross-validation. Full results for all traits in UKB and TEDS indicate consistency across methods, with no trait performing unexpectedly well or poorly on any single method (S6–S9 Tables; S4–S7 Figs).

## Comparison of polygenic scoring methods

A pairwise comparison of polygenic scoring methods was performed for each method (pT+clump, lassosum, PRScs, SBLUP, SBayesR, LDpred1, LDpred2, DBSLMM, All) and each model (10-fold cross validation, multi-PRS, pseudovalidation and infinitesimal). Fig 3 shows the difference in correlation (R) within and between methods for UKB outcomes with 1KG reference panel, with p-values for significant differences calculated using the William's test results aggregated across outcomes. Full results for TEDS and UKB, and for both reference panels are given in S10–S13 Tables and S8 Fig, and by trait in S14–S17 Tables.

When using 10-fold cross validation to identify the optimal parameter, LDpred2, lassosum and PRScs provided the most predictive polygenic scores in the test sample on average, with a

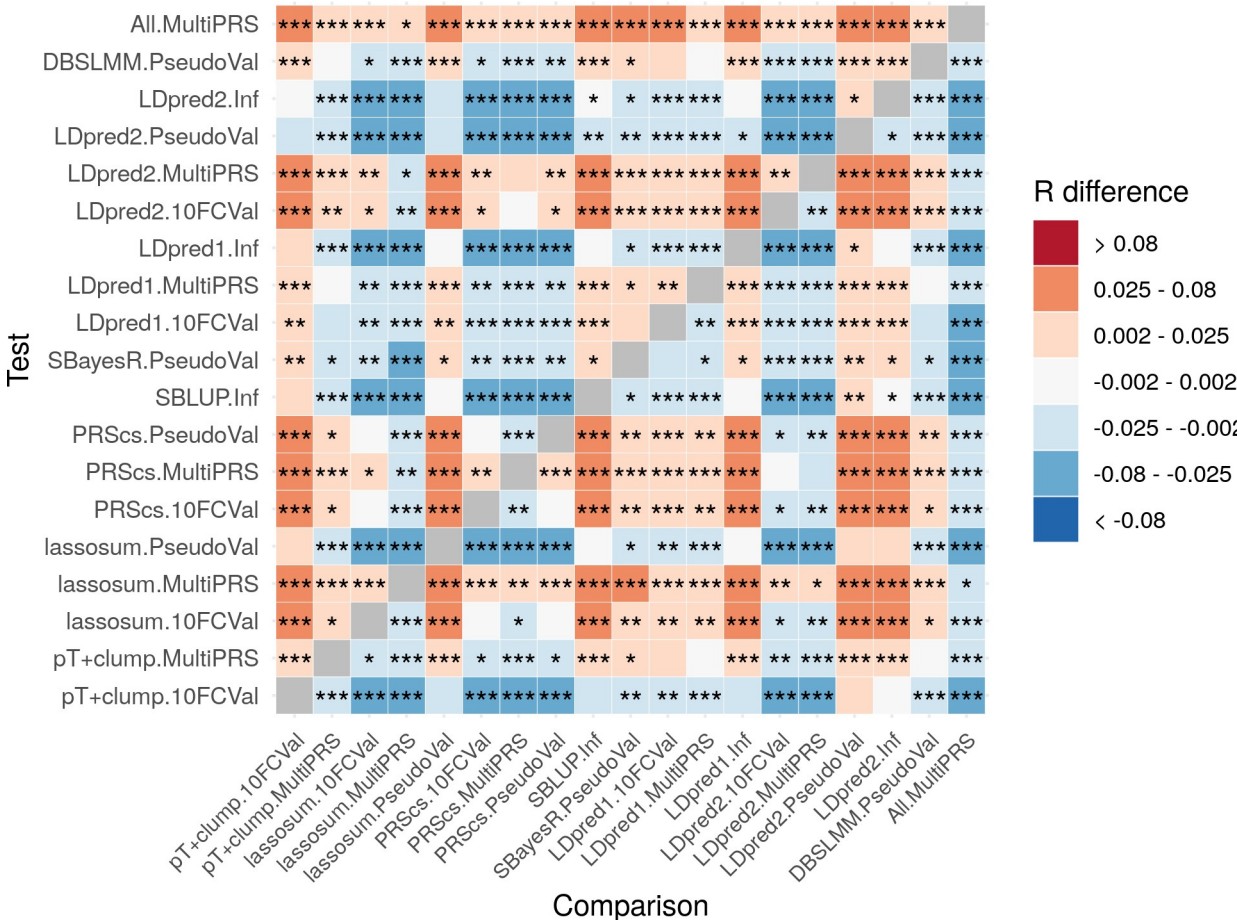

**Fig 3. Pairwise comparison between all methods, showing average test-set observed-expected correlation difference between all methods with significance value.** Correlation difference = Test correlation–Comparison correlation. Red/orange coloring indicates the Test method (shown on Y axis) performed better than the Comparison method (shown on X axis). Shows only results based on the UKB target sample when using the 1KG reference. * = p<0.05. ** = p<1×10⁻³. *** = p<1×10⁻⁶. P-values are two-sided. 10FCVal represents a single polygenic score based on the optimal parameter as identified using 10-fold cross-validation. Multi-PRS represents an elastic net model containing polygenic scores based on a range of parameters, with elastic net shrinkage parameters derived using 10-fold cross-validation. PseudoVal represents a single polygenic score based on the predicted optimal parameter as identified using pseudovalidation, which requires no tuning sample. Inf represents a single polygenic score based on the infinitesimal model, which requires no tuning sample.

16–18% relative improvement ($p<8\times10^{-16}$) over the 10-fold cross-validated pT+clump approach. When using 10-fold cross validation, on average LDpred2 provided a small but nominally significantly improved prediction over lassosum and PRScs (2%, p = 0.05).

Pseudovalidation and infinitesimal models do not require a tuning sample and their results are therefore described in parallel. The methods providing a pseudovalidation and/or infinitesimal approach include lassosum, PRScs, LDpred, LDpred2, SBLUP, DBSLMM and SBayesR. When using the smaller 1KG reference panel PRScs and DBSLMM performed the best on average, providing at least a 5% relative improvement ($p<2\times10^{-2}$) over other pseudovalidation approaches. The PRScs pseudovalidation approach provided a further significant improvement over DBSLMM, with an average relative improvement of 4% ($p = 4\times10^{-4}$). Furthermore, the PRScs pseudovalidation approach was on average only 3% (*p*-value = $6\times10^{-3}$) worse than the best polygenic score identified by 10-fold cross validation for any method. When using the larger UKB reference panel, the performance of SBayesR improved and was not significantly different to that of DBSLMM. The performance of lassosum pseudovalidation, the LDpred1 and LDpred2 infinitesimal models, SBLUP, LDpred2 pseudovalidation and SBayesR was variable across phenotypes, whereas the PRScs pseudovalidated polygenic score achieved near optimal predication compared to any method, and always performed better than the best pT+clump polygenic scores as identified by 10-fold cross validation. The performances of DBSLMM, and SBayesR when using the larger UKB reference were also relatively stable across phenotypes.

Modelling multiple polygenic scores based on multiple parameters using an elastic net consistently outperformed models containing the single best polygenic score as identified using 10-fold cross validation. The improvement was largest when using pT+clump polygenic scores (12% relative improvement, $p = 1\times10^{-21}$), but was also statistically significant for lassosum (6% relative improvement, $3\times10^{-15}$), PRScs (2% relative improvement, $p = 4\times10^{-5}$), LDpred1 (2% relative improvement, $p = 4\times10^{-5}$) and LDpred2 (2% relative improvement, $p = 3\times10^{-4}$ methods. On average, the 'All' method, combining polygenic scores across polygenic scoring methods provided a statistically significant improvement over the single best method (multi-PRS lassosum, 2% relative improvement, $p = 4\times10^{-3}$). Elastic net models using non-nested or dense *p*-value thresholds showed no improvement over the standard *p*-value thresholding approach (S18 and S19 Tables).

Convergence issues occurred for SBayesR for 4 of the 14 GWAS. In the latest version of the software implementing SBayesR (GCTB v2.03), developers have included a robust parameterization option which is automatically turned on when convergence issues are detected. We found that the robust parameterization resolved convergence issues, although the software had limited ability to detect convergence issues (S9 and S10 Figs). As a result, we recommend specifying the '--robust' option to force the robust parameterization, as this optimized SBayesR performance in most instances (S9 and S10 Figs). Results comparing SBayesR to other methods reported in this study were derived using the robust parameterization option.

The relative performance of all methods and modelling approaches was similar across low and high polygenicity phenotypes (S11 Fig). Infinitesimal model-based polygenic scores performed better for high polygenicity phenotypes. Estimates of polygenicity for each phenotype are shown in S20 Table.

Controlling for the first 20 genetic principal components did not affect the relative performance of polygenic scoring methods (S12 Fig).

## Runtime comparison

The runtime of methods to process GWAS summary statistics on chromosome 22 without parallel implementations varied substantially (S13 Fig). The methods (fastest to slowest) were

pt+clump (~3 seconds), DBSLMM and lassosum (~30 seconds), SBLUP (~1 minute), SBayesR and LDpred1 (~3–6 minutes), PRScs (~35 minutes), and LDpred2 (~50 minutes). The number of parameters tested by each method will influence the runtime. For example, using only one shrinkage parameter for PRScs will take 1/5 of time taken for PRScs to use 5 shrinkage parameters.

## Discussion

This study evaluated a range of polygenic scoring methods across phenotypes representing a range of genetic architectures and using reference and target sample genotypic data of different sample sizes. This study shows that, when a tuning sample is available to identify optimal parameters, more recently developed methods that do not perform LD-based clumping provide better prediction, with LDpred2, lassosum and PRScs providing a relative improvement of 16–18% compared to the pT+clump approach. When a tuning sample is not available, the optimal method for prediction was PRScs, with DBSLMM and SBayesR also performing well. Furthermore, the PRScs pseudovalidation performance was only 3% worse than the best polygenic scores identified by 10-fold cross validation for any other method. This study also shows that an elastic net model containing multiple polygenic scores based on a range of p-value thresholds or shrinkage parameters provides better prediction than the single best polygenic score as identified by 10-fold cross validation. Modelling multiple parameters increased prediction by 12% when using the pT+clump approach and 2–6% for polygenic scoring methods that model LD. Modelling polygenic scores from multiple methods provided a relative improvement of 1.7% in prediction over the single best method, though the additional computation time to perform all methods is substantial.

Our study highlighted the performance of SBayesR using default settings is highly variable across GWAS summary statistics due to convergence issues. However, convergence issues are avoided when the newly implemented robust parameterization option is specified.

These methods were evaluated within a reference-standardized framework and the results are likely to be generalizable to a range of settings, including a clinical setting. The improved transferability of prediction accuracy when using a reference-standardized approach enables prediction with a known accuracy for a single individual. This is an essential feature of any predictor as then its prediction can be appropriately considered in relation to other information about the individual. It is important to consider whether the reference-standardized approach impacts the predictive utility of the polygenic scores compared to those derived using target sample specific properties. The use of only HapMap3 variants is common for polygenic scoring methods as denser sets of variants increase the computational burden of the analysis and provide only incremental improvements in prediction [12]. However, denser sets of variants are ultimately likely to be of importance for optimizing the predictive utility of polygenic scores. The use of reference LD estimates instead of target sample-specific LD estimates is less likely to impact the predictive utility of polygenic scores. LD estimates are used to recapitulate LD structure in the GWAS discovery sample, and there should therefore be no advantage to using target sample specific LD estimates instead of reference sample LD estimates, unless the target sample better captures the LD structure in the GWAS discovery sample.

One major limitation of our study is that it was performed only in studies of European ancestry since GWAS of other ancestries have insufficient power for polygenic prediction. Polygenic scoring method comparisons in other ancestries or across ancestries will require substantial progress in diversifying genetic studies to non-European ancestry. In particular, it will be important to assess the impact of greater genetic diversity and weaker linkage disequilibrium in African ancestry populations. These studies are essential if polygenic risk scores are to be implemented in clinical care, to ensure equity of healthcare.

The clinical implementation of polygenic scores is at an early stage, and we identify five areas that still require further research. First, this study demonstrates that the reference-standardized approach provides reliable polygenic score estimates. However, the extent to which missing genetic variation within target sample data affects the prediction accuracy needs to be investigated. Furthermore, the extent to which prediction accuracy varies across individuals from different European ancestral populations needs to be assessed. Second, this study used the HapMap3 SNP list when deriving polygenic scores, building on previous research suggesting that these variants are reliably imputed and provide good coverage of the genome [20]. However, other sets of variants should be explored as denser coverage of the genome may improve prediction. Third, this study investigates polygenic scores based on a single discovery GWAS or phenotype. Previous research has shown that methods which combine evidence across multiple GWAS can improve prediction due to genetic correlation between traits [37–41]. Further research comparing the predictive utility of multi-trait polygenic prediction within a reference-standardized framework is required. Fourth, we present the reference standardized approach as a conceptual framework for implementing polygenic scores in a clinical setting. However, several additional issues will need to be addressed before they can be used in a clinical setting, such as assigning individuals to the optimal reference population, the presence of admixture, and translating relative polygenic scores into absolute terms. Finally, integration of functional genomic annotations has been shown to improve prediction over functionally agnostic polygenic scoring methods [42]. Comparison of functionally informed methods within a reference-standardized framework is also required.

In conclusion, this study performed a comprehensive comparison of GWAS summary statistic-based polygenic scoring methods within a reference-standardized framework using European ancestry studies. The results provide a useful resource for future research and endeavors to implement polygenic scores for individual-level prediction. All the code, rationale and results of this study are available on the GenoPred website (see Web Resources). This website will continue to document the evaluation of novel genotype-based prediction methods, providing a valuable community resource for education, research, and collaboration. Novel polygenic score methods can be rapidly tested against these standard methods to benchmark performance. This framework should be a valuable tool in the roadmap of moving polygenic risk scores from research studies to clinical implementation. Further investigation of methods providing genotype-based prediction within a reference-standardized framework is needed.

## Supporting information

**S1 Fig. Schematic diagram showing UKB was split into reference, training and testing samples.** A sample of UKB providing 50,000 observations for each phenotype was identified. The sample was then further split into training (80%) and testing (20%) samples. The training sample used 10-fold cross validation to identify the optimal polygenic scoring parameters and elastic net hyper-parameters. An independent sample of 10,000 European UKB participants was also created to as a reference for polygenic scoring.
(PNG)

**S2 Fig. Average test-set correlation between predicted and observed values across phenotypes.** Error bars indicate standard error of correlations for each method. Results are split by the target and reference genotypic data used. Results are 10FCVal bars represent a single polygenic score based on the optimal parameter as identified using 10-fold cross-validation. Multi-PRS bars represent an elastic net model containing polygenic scores based on a range of parameters, with elastic net shrinkage parameters derived using 10-fold cross-validation. PseudoVal bars represent a single polygenic score based on the predicted optimal parameter as

identified using pseudovalidation, which requires no tuning sample. Inf represents a single polygenic score based on the infinitesimal model, which requires no tuning sample.
(PNG)

**S3 Fig. Average test-set observed-expected correlation difference between the best pT+clump polygenic score and all other methods.** The average difference across phenotypes are shown as diamonds with error bars indicating the standard error, and the difference for each phenotype shown as transparent circles. Results are split by the target and reference genotypic data used. 10FCVal represents a single polygenic score based on the optimal parameter as identified using 10-fold cross-validation. Multi-PRS represents an elastic net model containing polygenic scores based on a range of parameters, with elastic net shrinkage parameters derived using 10-fold cross-validation. PseudoVal represents a single polygenic score based on the predicted optimal parameter as identified using pseudovalidation, which requires no tuning sample. Inf represents a single polygenic score based on the infinitesimal model, which requires no tuning sample.
(PNG)

**S4 Fig. Correlation between predicted and observed values for each phenotype in UKB when using the European subset of 1000 Genomes as the reference.** Error bars indicate standard errors. 10FCVal bars represent a single polygenic score based on the optimal parameter as identified using 10-fold cross-validation. Multi-PRS bars represent an elastic net model containing polygenic scores based on a range of parameters, with elastic net shrinkage parameters derived using 10-fold cross-validation. PseudoVal bars represent a single polygenic score based on the predicted optimal parameter as identified using pseudovalidation, which requires no tuning sample. Inf represents a single polygenic score based on the infinitesimal model, which requires no tuning sample.
(PNG)

**S5 Fig. Correlation between predicted and observed values for each phenotype in UKB when using an independent 10K subset of European UKB individuals as the reference.** Error bars indicate standard errors. 10FCVal bars represent a single polygenic score based on the optimal parameter as identified using 10-fold cross-validation. Multi-PRS bars represent an elastic net model containing polygenic scores based on a range of parameters, with elastic net shrinkage parameters derived using 10-fold cross-validation. PseudoVal bars represent a single polygenic score based on the predicted optimal parameter as identified using pseudovalidation, which requires no tuning sample. Inf represents a single polygenic score based on the infinitesimal model, which requires no tuning sample.
(PNG)

**S6 Fig. Correlation between predicted and observed values for each phenotype in TEDS when using the European subset of 1000 Genomes as the reference.** Error bars indicate standard errors. 10FCVal bars represent a single polygenic score based on the optimal parameter as identified using 10-fold cross-validation. Multi-PRS bars represent an elastic net model containing polygenic scores based on a range of parameters, with elastic net shrinkage parameters derived using 10-fold cross-validation. PseudoVal bars represent a single polygenic score based on the predicted optimal parameter as identified using pseudovalidation, which requires no tuning sample.1000G Reference. Inf represents a single polygenic score based on the infinitesimal model, which requires no tuning sample.
(PNG)

**S7 Fig. Correlation between predicted and observed values for each phenotype in TEDS when using an independent 10K subset of European UKB individuals as the reference.**

Error bars indicate standard errors. 10FCVal bars represent a single polygenic score based on the optimal parameter as identified using 10-fold cross-validation. Multi-PRS bars represent an elastic net model containing polygenic scores based on a range of parameters, with elastic net shrinkage parameters derived using 10-fold cross-validation. PseudoVal bars represent a single polygenic score based on the predicted optimal parameter as identified using pseudovalidation, which requires no tuning sample. Inf represents a single polygenic score based on the infinitesimal model, which requires no tuning sample.
(PNG)

**S8 Fig. Average test-set observed-expected correlation difference between all methods with significance value.** Correlation difference = Test correlation–Reference correlation. Shows only results based on the UKB target sample when using the 1KG reference as other results were highly concordant. $^{*}$ = $p<0.05$. $^{**}$ = $p<1\times10^{-3}$. $^{***}$ = $p<1\times10^{-6}$. P-values are one-sided. 10FCVal corresponds to a single polygenic score based on the optimal parameter as identified using 10-fold cross-validation. Multi-PRS corresponds to an elastic net model containing polygenic scores based on a range of parameters, with elastic net shrinkage parameters derived using 10-fold cross-validation. PseudoVal corresponds to a single polygenic score based on the predicted optimal parameter as identified using pseudovalidation, which requires no tuning sample. Inf represents a single polygenic score based on the infinitesimal model, which requires no tuning sample.
(PNG)

**S9 Fig. Correlation between predicted and observed values across phenotypes in UKB for SBayesR polygenic scores derived using different reference samples and different GWAS processing procedures.** 1KG indicates the reference sample was the European subset of 1000 Genomes. UKB indicates the reference sample was an independent 10K subset of European UKB individuals. GCTB indicates the reference was the GCTB-provided reference data based on a non-independent 50K subset of European UKB individuals. The colour of the bars indicates the version of GCTB used when running SBayesR and which settings were used. Default indicates that default settings were used. P<0.4 indicates only variants with a GWAS p-value <0.4 were retained. Robust indicates that the—robust parameter was specified, forcing robust parameterisation.
(PNG)

**S10 Fig. Correlation between predicted and observed values across phenotypes in TEDS for SBayesR polygenic scores derived using different reference samples and different GWAS processing procedures.** 1KG indicates the reference sample was the European subset of 1000 Genomes. UKB indicates the reference sample was an independent 10K subset of European UKB individuals. GCTB indicates the reference was the GCTB-provided reference data based on a non-independent 50K subset of European UKB individuals. The colour of the bars indicates the version of GCTB used when running SBayesR and which settings were used. Default indicates that default settings were used. P<0.4 indicates only variants with a GWAS p-value <0.4 were retained. Robust indicates that the—robust parameter was specified, forcing robust parameterisation.
(PNG)

**S11 Fig. Comparison of methods across high and low polygenicity outcomes in UKB target sample using 1KG reference.** Figure shows average test-set observed-expected correlation difference between the best pT+clump polygenic score and all other methods. The average difference across phenotypes are shown as diamonds with error bars indicating the standard error, and the difference for each phenotype shown as transparent circles. 10FCVal represents a

single polygenic score based on the optimal parameter as identified using 10-fold cross-validation. Multi-PRS represents an elastic net model containing polygenic scores based on a range of parameters, with elastic net shrinkage parameters derived using 10-fold cross-validation. PseudoVal represents a single polygenic score based on the predicted optimal parameter as identified using pseudovalidation, which requires no tuning sample. Inf represents a single polygenic score based on the infinitesimal model, which requires no tuning sample.
(PNG)

**S12 Fig. Comparison of methods after controlling for genetic principal components in UKB target sample using 1KG reference.** Figure shows average test-set observed-expected correlation difference between the best pT+clump polygenic score and all other methods. The average difference across phenotypes are shown as diamonds with error bars indicating the standard error, and the difference for each phenotype shown as transparent circles. 10FCVal represents a single polygenic score based on the optimal parameter as identified using 10-fold cross-validation. Multi-PRS represents an elastic net model containing polygenic scores based on a range of parameters, with elastic net shrinkage parameters derived using 10-fold cross-validation. PseudoVal represents a single polygenic score based on the predicted optimal parameter as identified using pseudovalidation, which requires no tuning sample. Inf represents a single polygenic score based on the infinitesimal model, which requires no tuning sample.
(PNG)

**S13 Fig. Runtime for each polygenic scoring method using genetic variants on chromosome 22.** No parallel implementations were used.
(PNG)

**S1 Table. Descriptive statistics for GWAS summary statistics used to predict target sample phenotypes.**
(XLSX)

**S2 Table. Average performance in UKB with 1KG reference.**
(XLSX)

**S3 Table. Average performance in UKB with UKB reference.**
(XLSX)

**S4 Table. Average performance in TEDS with 1KG reference.**
(XLSX)

**S5 Table. Average performance in TEDS with UKB reference.**
(XLSX)

**S6 Table. Polygenic prediction in UKB using EUR 1KG reference.**
(XLSX)

**S7 Table. Polygenic prediction in UKB using EUR 10K UKB reference.**
(XLSX)

**S8 Table. Polygenic prediction in TEDS using EUR 1KG reference.**
(XLSX)

**S9 Table. Polygenic prediction in TEDS using EUR 10K UKB reference.**
(XLSX)

**S10 Table. Average difference between methods in UKB with 1KG reference.**
(XLSX)

**S11 Table. Average difference between methods in UKB with UKB reference.**
(XLSX)

**S12 Table. Average difference between methods in TEDS with 1KG reference.**
(XLSX)

**S13 Table. Average difference between methods in TEDS with UKB reference.**
(XLSX)

**S14 Table. Difference between methods in UKB with 1KG reference.**
(XLSX)

**S15 Table. Difference between methods in UKB with UKB reference.**
(XLSX)

**S16 Table. Difference between methods in TEDS with 1KG reference.**
(XLSX)

**S17 Table. Difference between methods in TEDS with UKB reference.**
(XLSX)

**S18 Table. Correlation between pT+clump model predictions and observed values in UK Biobank.**
(XLSX)

**S19 Table. Correlation between pT+clump model predictions and observed values in TEDS.**
(XLSX)

**S20 Table. AVENGEME Estimates of Polygenicity.**
(XLSX)

**S1 Supplementary Text. Provides description of outcome definitions in UKB and TEDS, methodology for estimating polygenicity, and SBayesR sensitivity analyses.**
(DOCX)

## Acknowledgments

We thank Luke Lloyd-Jones and Jian Zeng for their advice on SBayesR using GCTB, Tian Ge on using PRScs, and Florian Privé on using LDpred2. We thank Paul O'Reilly and Sam Choi for useful discussion.

The authors acknowledge use of the research computing facility at King's College London, Rosalind (https://rosalind.kcl.ac.uk), which is delivered in partnership with the NIHR Maudsley BRC, and part-funded by capital equipment grants from the Maudsley Charity (award 980) and Guy's & St. Thomas' Charity (TR130505). The views expressed are those of the authors and not necessarily those of the NHS, the NIHR or the Department of Health and Social Care. We thank the research participants and employees of 23andMe for making the work regarding Depression possible.

UKB: This research was conducted under UK Biobank application 18177.

TEDS: We gratefully acknowledge the ongoing contribution of the participants in TEDS and their families. TEDS is supported by UK Medical Research Council Program Grant MR/M021475/1 (and previously Grant G0901245) (to Robert Plomin (R.P).), with additional

support from National Institutes of Health Grant AG046938. The research leading to these results has also received funding from the European Research Council under the European Union's Seventh Framework Programme (FP7/2007-2013) ERC Grant Agreement 295366. R. P. is supported by Medical Research Council Research Professor Award G19/2. KR is supported by a Sir Henry Wellcome Postdoctoral Fellowship (213514/Z/18/Z).

## Web resources

- LDSC HapMap 3 SNP-list: https://data.broadinstitute.org/alkesgroup/LDSCORE/w_hm3.snplist.bz2

- LDSC Munge Sumstats: https://github.com/bulik/ldsc/blob/master/munge_sumstats.py

- GCTB LD matrices: https://zenodo.org/record/3350914

- Impute.me: https://impute.me/

- GenoPred: https://opain.github.io/GenoPred

## Author Contributions

**Conceptualization:** Oliver Pain, Lasse Folkersen, Cathryn M. Lewis.

**Data curation:** Kylie P. Glanville, Saskia P. Hagenaars, Saskia Selzam, Anna E. Fürtjes, Héléna A. Gaspar, Jonathan R. I. Coleman, Kaili Rimfeld, Gerome Breen, Robert Plomin.

**Formal analysis:** Oliver Pain.

**Funding acquisition:** Cathryn M. Lewis.

**Investigation:** Oliver Pain.

**Methodology:** Oliver Pain, Lasse Folkersen.

**Project administration:** Cathryn M. Lewis.

**Resources:** Kylie P. Glanville, Saskia P. Hagenaars, Saskia Selzam, Anna E. Fürtjes, Héléna A. Gaspar, Jonathan R. I. Coleman, Kaili Rimfeld, Gerome Breen, Robert Plomin, Lasse Folkersen.

**Software:** Oliver Pain.

**Supervision:** Cathryn M. Lewis.

**Visualization:** Oliver Pain.

**Writing – original draft:** Oliver Pain.

**Writing – review & editing:** Kylie P. Glanville, Saskia Selzam, Anna E. Fürtjes, Jonathan R. I. Coleman, Kaili Rimfeld, Gerome Breen, Robert Plomin, Lasse Folkersen, Cathryn M. Lewis.

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
