## [Decision Letter · Decision Letter 0]

25 Oct 2020

Dear Dr Pain,

Thank you very much for submitting your Research Article entitled 'Evaluation of Polygenic Prediction Methodology within a Reference-Standardized Framework' to PLOS Genetics. Your manuscript was fully evaluated at the editorial level and by independent peer reviewers. As you can see, the three reviewers engaged constructively with the paper and made good comments. Based on these reviews, we would welcome a revised version of the manuscript, while not being able to promise, of course, publication at that time.

Reading through the reviews, a specific comment that occurred to me before I sent this paper out for review and that was confirmed by reviewers is the need to report more standard metrics for binary outcomes (AUC/OR per SD). I mention this because one of my first reaction was to line up these results with my personal intuition, and while I appreciate your argument that a corresponding value can be estimated in theory, it remains a hurdle. So that would be a meaningful improvement.

A reviewer noted that not all existing methods are presented but that is understandable of course- there are simply too many. As an editor, I will not require that you extend a lot the set of methods included but please try to consider that comment if you can. On the other hand, a small number of additional disease traits (breast cancer for example?) would be helpful. The cancer space is of particular interest and should be highlighted.

More generally, the reviewers have made many suggestions for improvements and additional comparisons.  We won't expect you to fully meet every such request but we would like to see some substantial improvements in a revised version - we leave to you to choose the most important additions that can be made to improve the manuscript within a reasonable time frame.  Your response letter should address all the points made by each reviewer and justify the decisions that you made, as well as a description of the changes you have made in the manuscript.

If you decide to revise the manuscript for further consideration at PLOS Genetics, please aim to resubmit within the next 60 days, unless it will take extra time to address the concerns of the reviewers, in which case we would appreciate an expected resubmission date by email to plosgenetics@plos.org.

[LINK]

We are sorry that we cannot be more positive about your manuscript at this stage. Please do not hesitate to contact us if you have any concerns or questions.

Yours sincerely,

Vincent Plagnol

Associate Editor

PLOS Genetics

David Balding

Section Editor: Methods

PLOS Genetics

Reviewer's Responses to Questions

**Comments to the Authors:**

Reviewer #1: In this report, Pain et al. set out to benchmark various polygenic score algorithms as well as two reference panels across a range of 13 traits. They determine that on average: (i) all scores outperformed pruning and thresholding based approaches; (ii) PRScs delivered performance comparable to other approaches but dit not require a tuning set; (iii) use of a polygenic score based on an elastic net tended to outperform other strategies.

I do think that this approach will be of interest to the general genetics community.

* Please clarify - multiPRS for each algorithms are based on combining multiple scores derived from a single algorithm?

* Can score performance be improved using an elastic net of scores using multiple different algorithms do even better?

* I think correlation is used as a surrogate for accuracy, but I wonder if additional metrics should also be reported (e.g. a measure of discrimination for binary phenotypes or OR/SD commonly used), can get ideas from the Wand et al. preprint one polygenic score reporting standards from ClinGen

* For P&T, what r2 value(s) were used?

* Can authors comment on general principle of being able to predict which approach might work the best based on heritability, polygenicity, etc.? e.g. autoimmune diseases have a very different architecture so could be optimized using a different approach.

* For lassosum, is it appropriate to have a set lambda (as opposed to letting the software choose one as done in glimnet R package)?

* I found Figure 3 quite confusing and nonintuitive, would encourage an alternate presentation style.

* Please clarify how scores will be shared with research community.

Reviewer #2: This manuscript sets out to compare several methods for deriving polygenic

scores, across 13 phenotypes, in the UK Biobank and TEDS study. Several tuning

strategies are compared (10-fold cross-validation, pseudovalidation). With the

large number of polygenic score methods today, it is useful for the human

genetics community to benchmark which approaches are more promising.

Overall, this manuscript is a timely and useful addition to the literature. At

the same time, there are many methods being compared across different phenotypes

and often I found it difficult to follow the logic of the analysis and what

message each key result was trying to convey. In addition, some of the key

concepts (e.g., what pseudovalidation means for each method) aren't explained,

making the results hard to understand, especially for readers who aren't

experts in PRS.

Major comments:

* It's not clear whether the terminology of 'reference standardized approach'

narrowly refers to the use of the same LD reference panels and HapMap3 SNPs to

enable consistent evaluation of methods, or to a wider conceptual framework for

precision medicine; this really needs clarification. If it's only the former,

then it's simply good scientific practice to benchmark tools consistently and is

fine as is (with clarification). But if it's the latter (e.g., as hinted in

Discussion pg 22, line 453), then it should be acknowledged that this is only

the first step in such an endeavour and more complex issues will arise, e.g.,

matching the target sample to the best reference sample, admixture, absolute

risk and calibration, etc.

* What's the reasoning behind limiting the sample size to n=50,000 for all

phenotypes?

* Figure 1: What is the difference between the boxes 'Pre-imputed genotype data'

and 'Observed genome-wide genotype data'?

* An illustration of the actual study design (e.g., the split of UKB into

various reference/tuning/testing subsets etc, and sample sizes in each) would be

very useful as a supplementary figure, I found it difficult to follow which

dataset went into which analysis.

* Can you briefly explain pseudovalidation, and why you consider SBLUP/SBayesR

as methods that don't require cross-validation? LDpred has a single parameter

too, yet it was used in cross-validation.

* For the summary stats, is any MAF cutoff used, and are the same exact SNPs

used by all methods across all summary stats? e.g., LDpred v1 by default filters

out SNPs with MAF<1% and/or palindromic SNPs, but other methods may not.

* For the MultiPRS, it's not clear what kind of PRS went into it. Was it only

P+T based PRS or others as well? How many scores?

* Fig 2B, the caption and legend mention diamonds, but the actual figure shows

little stars, are they referring to the same thing?

* More of a comment / Discussion item, the cross-validation (including the

MultiPRS) will be affected by the number of cases in the training set, which is

quite small for some phenotypes in the UKB (e.g., multiple sclerosis), but isn't

really an inherent 'biological' feature of that disease/phenotype but more of a

UKB limitation.

* Figure 3: It's not clear what this figure is trying to convey. Does Reference

mean the n=10,000 UKB dataset? This should be explained in the figure or the

caption. Also the caption says 'For columns, red/orange coloring indicates the

Test method performed better than the Test method (horizontal)', do you mean

'performed better than the Reference method'?

Minor comments:

* For the UK Biobank, does 'European ancestry' here mean the British White

subset or is it a wider definition?

* pg 15, line 312 'logistic regression was used...', it's not clear why this is

under the 'Estimating prediction accuracy' subsection, and how does it apply to

these PRS methods? Most of the PRS methods assume a linear regression model even

for binary outcomes.

* Which version of LDpred was used?

* pg 5, line 63: 'Although genetic information is used to predict rare Mendelian

genetic disorders' -> most of the time rare Mendelian genetic disorders don't

need prediction since they manifest early in life, are highly penetrant, and

testing usually occurs after clinical presentation and not before. Perhaps

BRCA1/2 is the one standout example you can use here.

* pg 6, line 98: 'Effect sizes estimated in a GWAS are typically larger than

they would be in an independent sample due to overfitting or winner’s curse'.

This text could use some clarification as it could confuse readers; in standard

GWAS, there is no overfitting (one predictor with thousands of samples) but due

to p-value selection there is a winner's curse which leads to inflation of

effect sizes for the statistically significant SNPs. In contrast, in

multivariable models (e.g., BLUP or lasso) there can be overfitting since the

number of SNPs is typically much larger than the sample size.

* It's quite confusing to call one result '10FCVal' and another 'MultiPRS' when

the MultiPRS also used 10-fold cross-validation.

* pg 24, line 497, much more salient examples of MultiPRS (aka metaGRS)

improving predictive power over single PRS are

https://www.onlinejacc.org/content/72/16

https://www.nature.com/articles/s41467-019-13848-1

* Table 2: It's not quite right to say that e.g., LDpred needs only 'direction

of effect' as variant level data. LDpred v1 uses the (two-sided) p-value to

back-calculate the unsigned z-statistic, and it needs the effect direction to be

able to infer the sign of the original (signed) z-statistic. So it basically

needs both beta and the p-value.

* Discussion, pg 23 'Then prediction is the aim and inference is not of

interest...': this paragraph is confusing. Adjusting PRS for PCs doesn't affect

whether prediction is the aim or not, but affects interpretation. If there is

substantial population stratification in the GWAS and/or target population that

is also correlated with the phenotype of interest, the analysis needs to be very

careful due to potential for confounding. Arguably, we don't want PRS for T2D

being good at prediction just because they predict people's ancestry which is

confounded with socioeconomic status. It does get rather blurry for phenotypes

like educational attainment.

Reviewer #3: Pain et al present a well written and a very timely benchmark study for polygenic risk score methods. In recent years a number of different methods have been proposed to derive polygenic risk scores based on GWAS summary statistics, and almost every paper finds that their proposed method is somehow the “best”. As a user, it is therefore hard to understand what method to use for a given dataset. Therefore benchmark studies like these are particularly timely and important. However, I feel that the benchmarks provided here do not necessarily provide a full picture of the field. First, not all published methods are considered in the benchmark, although some of the most commonly used are included. Second, only about a dozen phenotypes (diseases and traits) are used for the benchmark, most of which are known to be highly polygenic. The results may therefore not generalise well for less polygenic traits, such as some types of cancer, metabolic measurements, and gene expression values. Besides these issues, I generally enjoyed the paper, and I really liked the accompanying GenoPred webpage. I list these and other comments below.

1. A number of methods have been published in recent years, which may be of interest. E.g. AnnoPred (Hu et al., PLoS Comp Biol 2017), SCT (Privé et al., AJHG 2019), NPS (Chun et al., AJHG 2020), and DBSLMM (Yang and Zhou, AJHG 2020). I realize many of these have only been published recently, but I would appreciate it if the authors included at least a couple of these in their comparison. Other methods that have been proposed, but are at the pre-print stage, are LDpred2 (Privé et al., bioRxiv), ldpred-funct (Marquez-Luna et al., bioRxiv 2020), MegaPRS (Zhang et al., bioRxiv 2020). Although it would be nice to include some of these in the comparison, I do not really expect the authors to do that, since these are not published yet.

2. Most or all of the traits/diseases considered here are highly polygenic. I recommend including a couple of more simple traits, as I am curious to understand what methods can predict those. These include blood and metabolite measurements, prostate cancer, breast cancer, or something like hair color, etc. Alternatively, you can examine the performance on different genetic architectures using simulations.

3. In my experience, QCing the GWAS summary statistics and the LD reference is really important in practice, especially for MCMC approaches such as PRScs, LDpred, and SBayesR. It is unclear to me how robust the different softwares are to data issues. I know some software provide a set of “good” variants, such as PRScs and LDpred2, for which they also provide a LD reference. Other software requires the user to QC the data beforehand. I would be interested in understanding what methods are more robust to data issues, such as poorly tagged LD, vastly different sample sizes for variants, duplicated variants, etc. Perhaps you can add a simulation where you perturb the data a bit, or at least provide an overview of how different methods deal with these issues.

4. If some methods are more prone to population stratification, adjusting for PCs can affect the relative results. I would be interested in seeing the relative comparison between methods before and after adjusting for PCs.

5. Can you please provide some statistics for the running times of the different methods.

6. Regarding the SBayesR, the results look weird as they suggest much worse performance than reported by the recent bioRxiv publications listed in point 1, and shown in Ni et al. (medRxiv 2020). Can you please elaborate on why this is the case.

7. 1KG is likely too small of an LD-reference for LDpred, SBLUP or SBayesR, which can affect relative performance. (As far as I can see, it is also not recommended.) Also, SBLUP and LDpred-inf are two different implementations of the same method/idea (see e.g. Ni et al. (medRxiv 2020)).

**Have all data underlying the figures and results presented in the manuscript been provided?**

Reviewer #1: Yes

Reviewer #2: Yes

Reviewer #3: Yes

PLOS authors have the option to publish the peer review history of their article (what does this mean?). If published, this will include your full peer review and any attached files.

Reviewer #1: No

Reviewer #2: No

Reviewer #3: No

---

## [Decision Letter · Decision Letter 1]

28 Mar 2021

Dear Dr Pain,

We are pleased to inform you that your manuscript entitled "Evaluation of Polygenic Prediction Methodology within a Reference-Standardized Framework" has been editorially accepted for publication in PLOS Genetics. Congratulations!  There are some comments from reviewers below that we ask you to address in preparing your final submission, but the editors do not require specific changes.

Yours sincerely,

Vincent Plagnol

Associate Editor

PLOS Genetics

David Balding

Section Editor: Methods

PLOS Genetics

Comments from the reviewers (if applicable)

Reviewer #1: I appreciate authors' efforts to improve the clarity of this manuscript.

Other metrics of potential interest would be calibration of various risk models (as recommended by recent polygenic score standards in Nature) and OR for top X versus median %, but I view these edits as discretionary

Reviewer #2: I thank the authors for their revised submission. In particular, the addition of LDpred2 is important since it is likely the best off-the-shelf PRS tool currently available.

All of my previous comments have been addressed and I do not have any further comments.

Reviewer #3: I would like to thank the authors for addressing all of my comments. The only concern that I still have is the discrepancy in the PRS performance between this paper and at least two other papers already available as preprints, namely Kulm et al. (https://www.medrxiv.org/content/10.1101/2020.04.06.20055574v2) and Ni et al. (https://www.medrxiv.org/content/10.1101/2020.09.10.20192310v1). I do understand that there are many possible reasons for this, and you do mention some possible reasons. However, as the three methods PRScs, LDpred1/2, SBayesR, are actually all quite similar in theory (all implement on a Gibbs sampler under different effect prior distribution), it is noteworthy that their results are so different. For these methods to work, the Gibbs sampler has to converge, which it can only do if the input data behaves. Hence, these discrepancies highlight the importance of QCing the GWAS summary statistics properly, using a good LD reference, and so forth. Maybe this is in part due to this QC process is not well described in the original papers, making it harder to use. These discrepancies may also be explained by implementation that are not sufficiently robust, Perhaps you can address these issues further in the discussion.

**Have all data underlying the figures and results presented in the manuscript been provided?**

Reviewer #1: Yes

Reviewer #2: Yes

Reviewer #3: Yes

PLOS authors have the option to publish the peer review history of their article (what does this mean?). If published, this will include your full peer review and any attached files.

Reviewer #1: No

Reviewer #2: No

Reviewer #3: No

**Data Deposition**

http://datadryad.org/submit?journalID=pgenetics&manu=PGENETICS-D-20-01242R1

**Press Queries**

---

## [Editor Report · Acceptance letter]

20 Apr 2021

PGENETICS-D-20-01242R1 

Evaluation of Polygenic Prediction Methodology within a Reference-Standardized Framework 

Dear Dr Pain, 

We are pleased to inform you that your manuscript entitled "Evaluation of Polygenic Prediction Methodology within a Reference-Standardized Framework" has been formally accepted for publication in PLOS Genetics! Your manuscript is now with our production department and you will be notified of the publication date in due course.

With kind regards,

Andrea Szabo

PLOS Genetics

On behalf of:
